# Open-Ended Coaxial Probe for Effective Reconstruction of Biopsy-Excised Tissues’ Dielectric Properties

**DOI:** 10.3390/s24072160

**Published:** 2024-03-28

**Authors:** Eliana Canicattì, Nunzia Fontana, Sami Barmada, Agostino Monorchio

**Affiliations:** 1FreeSpace Electromagnetic Technologies, 56121 Pisa, Italy; 2Department of Energy, Systems, Territory and Construction Engineering, University of Pisa, 56122 Pisa, Italy; sami.barmada@unipi.it; 3Department of Information Engineering, University of Pisa, 56122 Pisa, Italy; agostino.monorchio@unipi.it

**Keywords:** open-ended coaxial probe, millimeter-sized biopsy, fringing field, dielectric properties, virtual transmission line model (VTLM)

## Abstract

Dielectric characterization is extremely promising in medical contexts because it offers insights into the electromagnetic properties of biological tissues for the diagnosis of tumor diseases. This study introduces a promising approach to improve accuracy in the dielectric characterization of millimeter-sized biopsies based on the use of a customized electromagnetic characterization system by adopting a coated open-ended coaxial probe. Our approach aims to accelerate biopsy analysis without sample manipulation. Through comprehensive numerical simulations and experiments, we evaluated the effectiveness of a metal-coating system in comparison to a dielectric coating with the aim for replicating a real scenario: the use of a needle biopsy core with the tissue inside. The numerical analyses highlighted a substantial improvement in the reconstruction of the dielectric properties, particularly in managing the electric field distribution and mitigating fringing field effects. Experimental validation using bovine liver samples revealed highly accurate measurements, particularly in the real part of the permittivity, showing errors lower than 1% compared to the existing literature data. These results represent a significant advancement for the dielectric characterization of biopsy specimens in a rapid, precise, and non-invasive manner. This study underscores the robustness and reliability of our innovative approach, demonstrating the convergence of numerical analyses and empirical validation.

## 1. Introduction

Cancer, also known as malignant tumors, encompasses a range of diseases marked by uncontrollable and excessive cell growth, which can affect various parts of the body. Currently, hospitals use different biopsy methods and laboratory analytical techniques to characterize suspicious findings identified by conventional imaging. The gold standard technique for tissue excision is the core needle biopsy [1,2] followed by conventional pathology analysis. However, several limitations associated with those techniques have been identified and discussed [3,4]: the process for collecting tissues derived from biopsies may be subject to mistakes and imprecisions, which could potentially result in false negatives. Furthermore, the waiting period for pathology results is often long.

In this regard, technologies that adopt the dielectric characterization of tissues in the microwave range are widely investigated for innovative applications in cancer fighting. To this end, the adoption of devices with open-ended coaxial probes (OECPs) may be useful for tissue characterization, as they provide a valid support to conventional biopsies with the added advantage of real-time results [5,6,7]. These tools may have several clinical applications; they are characterized by simple usage, do not require sample manipulation, and are quick and inexpensive for tissue pre-classification. By measuring the frequency-dispersive dielectric characteristics, the characterization of the biological tissues under testing is achieved, in other words, to differentiate healthy tissue from tumorous tissue. In this regard, it is well established that the dielectric properties of healthy tissues and their tumor counterparts are different: the higher values characterizing dielectric parameters of cancerous tissues are primarily due to their higher average water contents [8,9,10,11].

Numerous studies provide evidence for the effectiveness of the OECPs in characterizing the dielectric properties of both healthy and cancerous tissues. These studies involve various types of tissues, including breast tissues [12,13,14,15], in vitro cell lines derived from both normal and tumorous breast tissues [16,17], colon tissues [18], and liver tissues [9]. Additionally, OECPs have been used in the identification of hepatic malignancies [19,20], aided by machine-learning techniques, as well as in the analysis of skin lesions [21]. Current probes present on the market have various limitations [7], such as requiring multiple probes for broadband characterization, high costs, and the complexity of the calibration process, which involves SOL (short, open, and load) systems. To overcome these challenges, the authors, in a previous work, optimized and developed [22] an OECP. This probe effectively combines mechanical practicality and the use of the so-called virtual transmission line model (VTLM) reconstruction algorithm.

However, the operational principle of the OECP assumes that the sample being examined has a semi-infinite size [7,23,24]. Nevertheless, biopsies can vary in size, and in certain instances, their small dimensions may not meet the aforementioned assumption, resulting in potentially inaccurate outcomes. Therefore, it is crucial to evaluate the minimum sample size required to ensure precise and reliable characterization.

Some studies in the literature focus on analyzing the detection volume of OECPs. Hagl et al. [25] conducted experiments using two commercial coaxial probes of different sizes. They used significant quantities of liquids (alcohol and deionized water) to emulate healthy and cancerous tissues, respectively. As stated in [26], the sensing volume of a coaxial probe is directly correlated with the internal radius of the outer conductor. This relationship was confirmed through experimental trials involving perforated Teflon™ blocks inside saline solutions. Furthermore, in [27], the authors investigated the variation in the probe penetration depth concerning the diameter when dealing with heterogeneous tissues. According to Martellosio et al. [28], they found that the mechanical restrictions caused by the size of the external probe made it impossible to use probes with diameters smaller than 5–6 mm. The researchers obtained these findings by conducting measurements on several materials of varying sample thicknesses and analyzing the differences in measurements with and without the presence of a metal plate beneath the material. 

Similarly, in previous research, we assessed the detection capability of our custom-designed OECP for characterizing biopsy samples. We extensively investigated the minimum detectable size of a biopsy sample, accounting for variations in their shapes. Initial evaluations indicated a minimum detectable spherical sample size of approximately 4 mm in diameter [29,30]. However, considering that biopsy tissues often have a cylindrical shape, we demonstrated the successful characterization of cylindrical biopsy samples measuring 2 mm (12 G) in diameter, achieving a reconstruction error of less than 9% in determining dielectric properties [31]. Nevertheless, when dealing with specimens of 1.8 mm (13 G) in diameter, we encountered a significant challenge, experiencing a reconstruction error exceeding 15% due to electric field dispersion. To address this limitation, we introduced a dielectric coating, effectively mitigating the field dispersion and enhancing the accuracy for characterizing smaller specimens. 

In this paper, we present the validation and verification of the impact of a metallic coating, comparing it with a dielectric coating, to enhance results and minimize reconstruction errors in the dielectric characterization of millimeter-sized biopsies. Our approach involves applying a metal coating to the probe–tissue region to mitigate the fringing field effect occurring in air region. We initially conducted a set of full-wave simulations with a numerical model implemented in CST Studio Suite 2019^®^ software [32]. We designed and analyzed numerical tissues with cylindrical shapes in a frequency range spanning from 10 MHz to 3 GHz. We assessed how the presence of the metal coating improved the dielectric characterization results. Subsequently, we compared these outcomes with those from our previous studies, where we utilized a PTFE dielectric coating. These numerical findings were then validated through laboratory experiments. We employed PLA dielectric supports, some of which were coated with special metallic paint, to simulate both the metallic coating and the PTFE coating. Inside these supports, we placed samples of bovine liver (No animals were mistreated for conducting the present research activity. The liver samples were obtained from a bovine liver slaughtered for food consumption purposes.) and measured their dielectric properties.

This paper is organized as follows: In Section 2, we describe our custom-designed electromagnetic characterization system used to determine dielectric parameters. The subsequent section presents numerical results obtained from numerical simulations mimicking healthy and cancerous tissues. Specifically, we present the results of the characterization both in the absence of a coating and with the presence of both dielectric and metallic coatings. The paper concludes with experimental results, an in-depth discussion, and conclusions.

## 2. Materials and Methods

### 2.1. Open-Ended Coaxial Probe Design

The OECP consists of a truncated section of a coaxial cable connected to a vector network analyzer (VNA N9918A FieldFox, Keysight, Santa Rosa, CA, USA) for acquiring the reflection coefficients (S_11_). These coefficients, once acquired, allow for the reconstruction of material dielectric properties in terms of complex permittivity using proper inversion algorithms based on an equivalent analytical probe model [33]. 

Our previous work involved the development of a customized OECP aimed at overcoming the defects and limits of the existing commercial ones (e.g., the need to acquire kits including one or more probes to conduct broadband characterizations, resulting in high costs, and the burdensome calibration phases with open/short/load-type systems). Hence, we designed an optimized OECP, achieving a good balance between its feasibility and ease of integration with a reconstruction algorithm (the VTLM) [22]. In particular, the algorithm is based on representing the OECP terminated on a material having given specified characteristics, as a transmission line in which the material is modeled as a virtual transmission line of length *L*, i.e., the same as the geometrical length of the probe. Finally, this equivalent line is terminated as an open circuit. By the analytical formulation of the equivalent transmission line, expressed as the equivalent admittance along the line, which is related to the permittivity of the medium to be reconstructed, and solving for the permittivity, the algorithm can be used to estimate the dielectric properties of the tested medium by resorting to a single measurement of the scattering parameter (S_11_), which is related to the input impedance of the line, at the VNA. In our previous OECP realization, by optimizing the system using the analytical theory of transmission lines, we ensured that the impedance at the line’s input was entirely reactive [34]. This entailed that to prevent probe radiation losses, the real part of the line’s input impedance became zero. To this aim, we applied the empirical criterion described in [22]. Consequently, the coaxial probe consists of an inner dielectric, Teflon™ (PTFE), which has a relative permittivity (ε′) of 2.1, and has an external diameter of 1.68 mm. This system enables the estimation of the dielectric properties of materials across a broad frequency spectrum (10 MHz–10 GHz). Moreover, it requires straightforward calibration using commonly available and two affordable materials, such as distilled water and air. 

First, to validate the proposed method, we performed full-wave numerical simulations using CST Studio Suite (Dassault Systèmes) [32]. These simulations spanned the frequency range [10 MHz–3 GHz] and involved designing two numerical models mimicking biological tissues (cortical bone and skin), accounting for their dispersive dielectric properties within this frequency range. 

We reconstructed the dielectric properties of the tissues under investigation by applying the optimized algorithm to the reflection coefficient results, as depicted in Figure 1. It is worth observing the good agreement between the reconstructed data and the literature data. Subsequently, the OECP prototype was built (Figure 2) and experimentally validated. To this aim, we have dielectrically characterized liquids with known dielectric properties, such as saline solutions, glycol, distilled water, and oil. 

Specifically, to have a comparison with data reported in the literature [33,35], we prepared a saline solution with 99.9% pure, Sigma-Aldrich (St. Louis, MO, USA), NaCl [36] and deionized water. The solution is produced at a concentration of 1 M (58.44 g/L) NaCl within appropriate sterile containers to avoid the risk of any contamination. Figure 3 shows the experimental measurements of the solution at a concentration of 1 M and the comparison with the literature data in the frequency range 500 MHz–3 GHz.

To quantify the discrepancy between the results gathered using our characterization system and the theoretical model, the following percentual error is used: (1)$$\%\;PE=\left|\frac{V_{Calculated}-V_{Theoretical}}{V_{Theoretical}}\right|\times 100$$
where *V*_*Theoretical*_ represents a commonly accepted value (according to the literature data). Conversely, *V*_*Calculated*_ represents a numerically or experimentally derived quantity; in this case, it represents the dielectric properties of the saline solution measured using our characterization method. The percentage error (%*PE*) compares an estimated value to the correct one and expresses the difference as a percentage. This statistic enables us to determine the error magnitude with respect to the actual value. 

The results of the *%PE* calculated for 1 M saline solution with respect to the literature data are reported in Figure 4. 

Subsequently, we performed additional validation measurements using a section of pigskin (Figure 2b). The results acquired were deemed to be satisfactory and revealed good agreement with those yielded by the numerical ones. The experimental findings are depicted in Figure 5a,b.

In Figure 5c,d, the percentage errors calculated between our experimental results and the theoretical dispersive model [37] are depicted. It is possible to observe a percentage error (*PE*) lower than 15%. We extensively tested the proposed characterization system using both experimental measurements and numerical simulations. This comprehensive approach demonstrated high precision in evaluating dielectric properties: the comparison between our experimental and numerical findings with theoretical data confirmed the reliability and precision of our system.

### 2.2. Coated Open-Ended Coaxial Probe

The OECP working principle is based on the interaction between the fringing fields (which are generated between the internal and external conductors of the transmission line) and the investigated material [38,39]. The fields fringe out from the probe’s aperture, and they interact with the material under testing, altering the reflection coefficient; thus, the reflected signal might be correlated with the relative permittivity. However, when the sample size is roughly comparable to the aperture diameter of the probe, the effect of spurious fringing fields at the edges of the outer conductor may occur in free space, altering the correct measurement of the reflection coefficient and, consequently, the estimation of the dielectric properties. 

To prevent or reduce this phenomenon and, hence, to increase the accuracy of the electromagnetic characterization of millimeter-sized biopsy specimens, our previous work suggested a solution that confines the fringe fields to the region between the probe tip and material interface [31]. To this aim, we have designed a properly coated system. The Teflon coating (relative permittivity ε′ = 2.1) was extended over the probe tip and used as holder for the biopsy sample. Consequently, the occurrence of field fringing at the termination has been reduced, allowing for optimal electric field confinement and achieving an improved reflection coefficient estimation. As a result, a more accurate reconstruction of the dielectric properties was achieved.

In our current study, we propose using a metallic coating as a containment system for the biopsy sample. This coating, potentially represented by the biopsy needle itself, enables direct contact with the probe aperture’s plane and ensures electrical continuity with the coaxial cable’s shield. This setup offers a twofold advantage by ensuring efficient electric field confinement and maintaining the biopsy enclosed within the metallic-coated needle; this simplifies the assessment of dielectric properties after the sample extraction, avoiding the need for additional sample manipulation.

The effect of the metallic coating has been numerically assessed by means of full-wave simulations using a finite element method (FEM) in CST Studio Suite (Dassault Systèmes), which provided the electric field distribution and reflection coefficients as results. The metallic coating that contains the tissue has been designed as a hollow cylinder with an external diameter of 2.1 mm and a thickness of 0.1 mm (Figure 6a,b). 

We delved into malignant liver tissue modeled as a cylinder. Their diameters of 2 mm (12 G) and 1.8 mm (13 G) were selected according to the commonly used biopsy needle sizes; a probe of 0.1 mm was inserted. To faithfully mimic the dielectric properties of malignant liver samples, we assigned to the numerical models the dielectric properties using a first-order Debye model [9] within the frequency range of interest. The simulations were carried out in the frequency range [10 MHz–3 GHz], obtaining the reflection coefficients, which were post-processed with the VLTM reconstruction algorithm. 

At first, simulations were conducted by placing the cancerous tissue in direct contact with the coaxial probe (Figure 7), without any additional coating, to establish a starting point for the investigation. Later, a metallic coating was applied to examine its functions in reducing the fringe effects of the electric field and improving the faithful reconstruction of dielectric properties.

## 3. Results and Discussion

### 3.1. Numerical Results

#### 3.1.1. Metallic-Coated Open-Ended Coaxial Probe

The dielectric properties were determined by applying our in-house reconstruction algorithm to numerically calculated reflection coefficients. Initially, we assessed the results obtained by placing the numerical liver tissue in free space directly in the probe-to-tissue aperture’s plane (Figure 7). Figure 8 depicts the results for cylindrical liver tissues with diameters of D = 1.8 mm and D = 2 mm. 

To assess the reliability and accuracy of these results, we calculated the percentage error (*PE*) using Equation (1) based on the chosen theoretical dispersion model [9]. This analysis was conducted at a frequency of 1 GHz. The results in Table 1 show that the percentage error consistently remains below 15% when compared to those of the literature’s dispersive model. However, a significant discrepancy in the percentage error becomes apparent when comparing the two cylindrical samples possessing different diameters. Thus, it can be concluded that the accuracy of the dielectric property’s reconstruction is significantly affected by the diameter of the sample under investigation. Indeed, for the numerical cylindrical liver tissue with a diameter of 1.8 mm (13 G), we observed a reconstruction error greater than 10%.

Examining the distribution of the electric field (Figure 9) reveals a significant phenomenon in the probe-to-tissue contact region of the sample. Owing to the fringing electric field, the effective area of the specimen is larger than its actual area [40]. When the sample size approaches the diameter of the probe’s aperture, as in the case of the 1.8 mm sample diameter, spurious fringing fields at the edges of the outer conductor of the probe may occur in free space, altering the correct measurement of the reflection coefficient and, consequently, the estimation of the dielectric properties. Subsequently, we evaluated the outcomes achieved by placing the numerical malignant liver within the metal coating. 

To address the observed effects of spurious fringing fields, particularly evident in cases involving smaller sample diameters, we explored their insertion into a metallic coating. This strategic placement aimed to establish electrical continuity with the outer conductor, thereby mitigating the impact of spurious fields in the probe aperture’s plane. So, we investigated a metallic coating for the worst-case result, namely, for the case of the D = 1.8 mm numerical cylindrical tissue. In Figure 10a,b, the results are depicted and compared with those of the theoretical dispersive model.

By applying Equation (1), we evaluated the PE over the entire frequency band. The results, depicted in Figure 10c,d, have revealed an overall percentage error (%*PE*) of less than 10% across the entire examined frequency range. These findings validate the system’s precision in characterizing the dielectric properties of biopsies, even when dealing with small specimens, such as those of merely 1.8 mm. When the sample was put inside the metallic coating, a significant improvement was observed compared to the case without the coating. 

#### 3.1.2. Comparison between Metallic-Coated and Teflon-Coated Open-Ended Coaxial Probes

In our previous work, we have proposed a solution to confine the fringe fields to the region between the probe tip and material interface [31]. To this aim, we have designed a coated system (Figure 11). This approach effectively mitigated the fringing field effect in the surrounding air, facilitating the optimal confinement of the electric field lines. Consequently, it enabled more precise measurements of the reflection coefficient, leading to an accurate reconstruction of the dielectric properties.

The lossless Teflon coating functioned as a cost-effective and disposable holder for the biopsy samples. Additionally, it facilitated faster reflection coefficient measurements by minimizing the need for frequent sample manipulation. In examining the electric field of the coated probe, the coating aids in conveying electric field lines to the sample under examination. Specifically, the use of the dielectric coating reduces the fringing field in the air, resulting in a more confined electric field within the tissue being tested.

An in-depth analysis of the electric field’s behavior under two distinct conditions, by employing metallic and dielectric coatings, demonstrated a noticeable confinement of the electric field lines, as illustrated in Figure 12a and Figure 12b, respectively.

According to [41], standing waves in the coaxial line are the main cause of the radiative fringing effects in the probe aperture’s plane. A fringing field occurs near the aperture of the sensor. Hence, a phase shift between the forward wave and the reflected wave occurs and produces the standing wave inside the coaxial line [42]. 

To quantify the impacts of such coatings on the accuracy of the dielectric property’s reconstruction, we assessed the percentage errors (%*PE*s) in reconstructing liver dielectric properties at a frequency of 1 GHz. The results are summarized in Table 2 as follows:

Both the metallic and dielectric coatings play distinct roles in reducing the effects of standing waves and fringing fields on coaxial lines, albeit through different mechanisms. When a metallic coating is applied, it can act as a shield, confining the electric field within the coaxial structure more effectively. This lowers the creation of unwanted spurious electric field lines and lowers the interference brought about by external electromagnetic fields. The conductivity of the metallic coating aids in directing and containing the field, preventing it from radiating into the surrounding environment, thus mitigating the fringing effect. However, it exhibits the drawback of reduced performance when assessing the conductivity of the sample. When the electric field lines converge on the material, a portion may penetrate the metal coating, modifying the actual measurement of the sample’s conductivity. On the other hand, the dielectric coating, such as Teflon (PTFE), minimizes reflections and standing waves. By applying a dielectric coating, the back surface currents on the outer conductor of the coaxial cable can be limited, reducing the formation of spurious electric field lines in the air. When a Teflon coating is applied to an OECP, it acts as an insulating layer. With this coating, the probe will have a higher impedance (electrical resistance), which will increase the accuracy of measurement and the ability to detect weak signals. The Teflon coating also helps to reduce the unwanted reflections that can cause measurement errors. 

### 3.2. Experimental Results

Besides numerical studies, we also performed measurements on fabricated prototypes. We exploited 3D printing technology to produce the coating for the coaxial cable, which serves as a housing system for the specimen sample. Specifically, a 3D printer with a PLA filament was used to produce cylindrical components made of the PLA material. Such cylinders had a diameter of 1.68 mm and a thickness of 0.2 mm (Figure 13). 

PLA (polylactic acid) exhibits a dielectric constant of 2.7 at a frequency of 1 MHz, together with a low loss tangent of 0.008 [43]. Thus, the PLA cylinder worked as a counterpart to the dielectric Teflon coating. On the other hand, to simulate the metallic coating, conductive spray paint was used to apply a metallic layer to the cylinder, thus replicating the characteristics of a metallic coating. Laboratory testing was performed using a digital multimeter to verify the performance of the conductive paint. Resistance measurements were conducted on both the standalone metalized cylinder and in conjunction with the OECP to determine their electrical continuity (Figure 14).

Experimental measurements were conducted by connecting the OECP to a calibrated VNA (N9918A FieldFox, Keysight, Santa Rosa, CA, USA) via a standard SMA connector. The reflection coefficient (S_11_) was specifically recorded within the 10 MHz–3 GHz microwave frequency range at over 1001 frequency points on a linear scale. Before proceeding with the electromagnetic characterization and analysis of the dielectric parameters, the probe was calibrated using air and distilled water [22]. The measurement equipment (VNA and OECP) was arranged in a temperature-controlled room (22 °C) to maintain the system’s stability and ensure high measurement accuracy. The analyses were performed on bovine liver samples that were precisely sectioned and positioned within 3D-printed supports.

Subsequently, the samples dielectric properties were assessed using the OECP, as depicted in Figure 15a,b. Following the data acquisition, the permittivity and conductivity were reconstructed in real time using the inversion algorithm, as illustrated in Figure 16a,b, respectively. Moreover, to validate the results, the percentage errors (%*PE*s) were evaluated using the theoretical dispersive model, as in reference [9].

As shown in Table 3, both the dielectric and metallic coatings exhibit high accuracy in detecting the dielectric properties of the 1.8 mm diameter liver sample. Both cases demonstrate highly accurate measurements of dielectric properties, mostly in terms of the real part of the permittivity, with errors of less than 1% when compared to the existing data in the literature.

The metallic coating exhibits a greater level of uncertainty when predicting the conductivity, revealing a slightly higher error rate in comparison to that of the dielectric coating. This result confirms the findings from full-wave simulations, which indicate that the conductivity of the metal coating affects the actual conductivity measurement of the sample. Furthermore, the discrepancy between the experimental and numerical results for PE (σ) in the case of the metallic coating can be attributed to several factors. For instance, it is possible that when the material was placed inside the coating, some paint came off and made contact with the sample. Another possibility is that the paint was not evenly distributed during the metallization process. In contrast, the dielectric coating exhibits a significantly lower percentage error in measuring both the dielectric permittivity and conductivity.

In this study, we explored a novel approach to enhance the dielectric characterization of millimeter-sized biopsies. The proposed systems could find application in the medical field to streamline the dielectric characterization process, eliminating the need for sample manipulation and speeding up the dielectric characterization as a support for traditional biopsy techniques. For this purpose, we investigated the use of both dielectric and metallic coatings. The dielectric coating, extended over the probe tip, is cost-effective and disposable. On the other hand, the metal coating was investigated as it could simulate the biopsy needle itself as a support containing the sample to be positioned directly in the probe opening’s plane. Both systems aim to accelerate reflection coefficient measurements, avoid sample manipulation, and enhance dielectric characterization. Although the metallic coating influences the characterization of the sample in terms of the conductivity, employing these coatings enabled the optimal confinement of the electric field distribution, mitigating the fringing effect in the air and thereby enhancing the precision in the reconstruction of dielectric properties.

## 4. Conclusions

In this research, we introduced a promising approach to enhance the precision of dielectric characterization in millimeter-sized biopsies with the aim of directly using the needle biopsy core. A coated OECP for the dielectric characterization of biopsy tissue samples has been investigated. The need to evaluate the coating performance is twofold: it allows for, at the same time, sample integrity and characterization from a dielectric point of view as a support for a pre-pathological analysis of malignant tissue recognition. 

Through comprehensive numerical simulations and laboratory experiments, we evaluated the effectiveness of a metal-coating system in comparison to a dielectric coating. Numerical assessments demonstrated a marked improvement in the dielectric property’s reconstruction, notably in effectively containing the electric field distribution and mitigating the effects of fringing fields. This improvement was further substantiated by empirical experiments conducted on bovine liver samples, affirming the practical efficacy and reliability of our proposed method. On one hand, the dielectric coating, extended over the probe tip, has proven to be a convenient and effective solution, allowing for dielectric characterization results with a percentage error lower than 1% in both the permittivity and dielectric conductivity. The authors believe that the result in terms of the dielectric permittivity’s reconstruction, as obtained from measurements, is very encouraging as the errors incurred both with metallic and dielectric coatings are essentially comparable: the presence of the metallic coating, under real conditions of use, leads to acceptable errors during measurements, thus reasonably allowing for the direct use of a needle with biopsy tissue inside. Furthermore, these findings indicate the need for additional exploration regarding the role of dielectric coatings in reconstructing dielectric permittivity properties. On the other hand, the metallic coating, although posing a challenge for influencing conductivity measurements, still represents a reasonable compromise in contributing to the improvement in the characterization of millimeter-sized biological materials. These findings signify a pivotal step forward in advancing biomedical diagnostic tools, offering a faster, more precise, and non-invasive approach to biopsy dielectric characterization as a support for traditional biopsies. Our study not only represents a significant milestone in refining dielectric characterization methodologies but also paves the way for future advancements, presenting the prospect of enhanced tools and procedures within the domain of medical diagnostics.

## Figures and Tables

**Figure 1 sensors-24-02160-f001:**
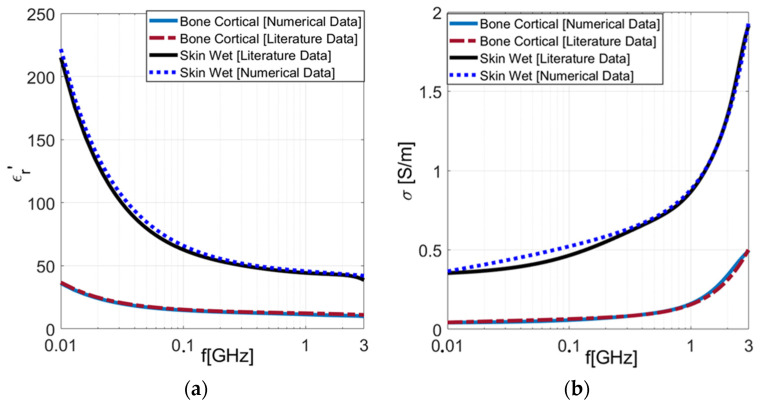
Numerical simulations: dielectric properties extracted by applying our in-house algorithm to numerical results. (**a**) Real part of dielectric permittivity. (**b**) Electrical conductivity.

**Figure 2 sensors-24-02160-f002:**
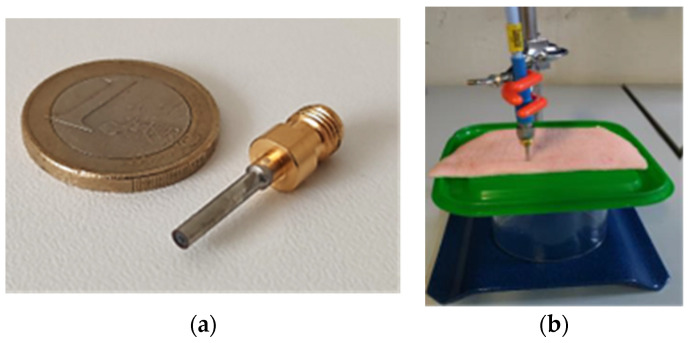
(**a**) Open-ended coaxial probe prototype; (**b**) pigskin measurements’ laboratory setup.

**Figure 3 sensors-24-02160-f003:**
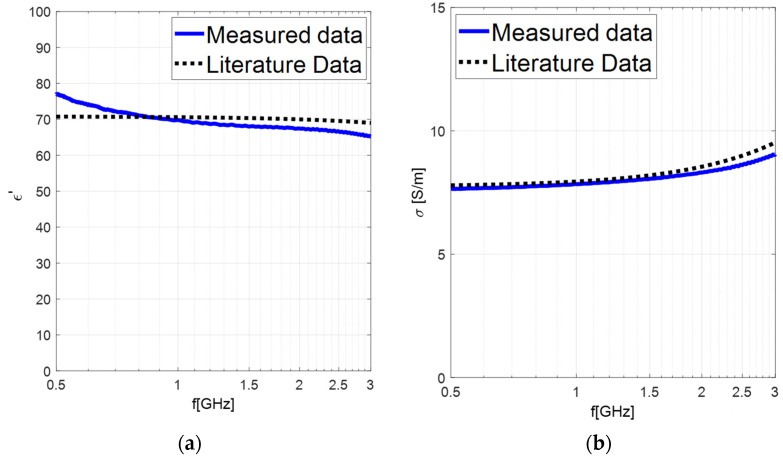
Experimental tissues’ dielectric properties extracted by applying our in-house algorithm to experimental results of 1 M saline solution in the frequency range 500 MHz–3 GHz. (**a**) Real part of dielectric permittivity. (**b**) Electrical conductivity.

**Figure 4 sensors-24-02160-f004:**
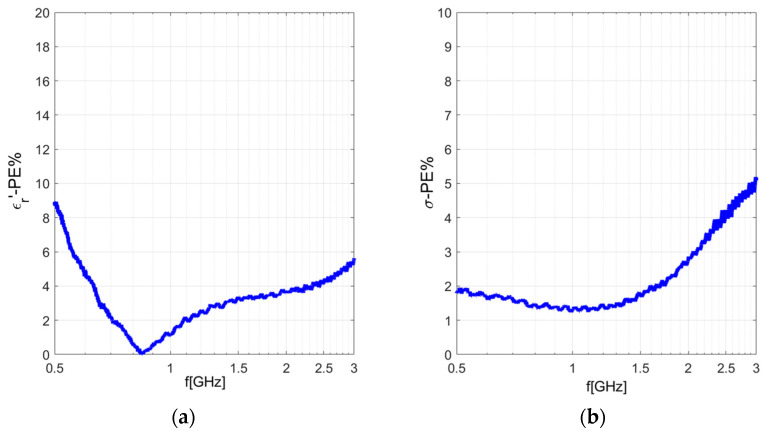
Percentage error calculated for 1 M saline solution with respect to the literature data [35] in the frequency range 500 MHz–3 GHz: (**a**) Real part of dielectric permittivity; (**b**) conductivity.

**Figure 5 sensors-24-02160-f005:**
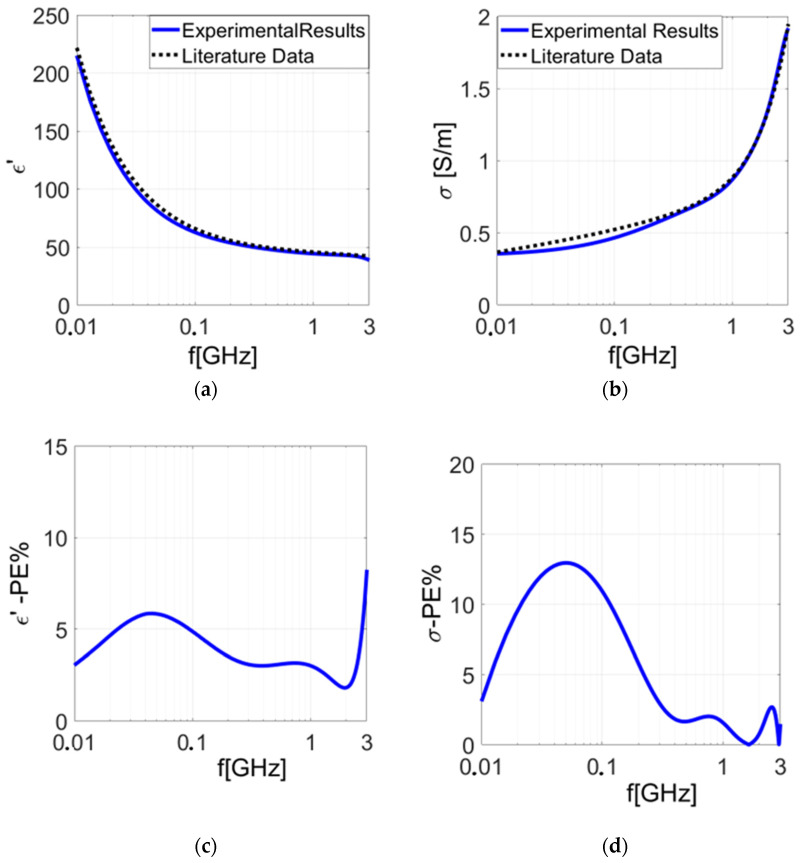
Dielectric properties measured on pigskin in the frequency range 10 MHz–3 GHz: (**a**) real part of dielectric permittivity; (**b**) electrical conductivity; (**c**) percentage error (*PE*) calculated for the real part of the dielectric permittivity; (**d**) percentage error (*PE*) calculated for the conductivity.

**Figure 6 sensors-24-02160-f006:**
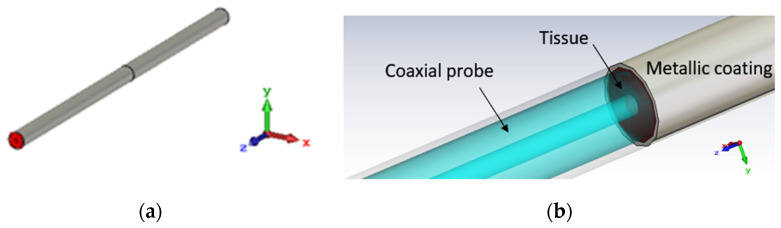
(**a**) A 3D numerical CST model of the open-ended coaxial probe and the metallic coating used as its extension and tissue housing: perspective view; (**b**) 3D numerical CST model of the open-ended coaxial probe and the metallic coating used as its extension: detail of the transition aperture’s plane between the coaxial transmission line and malignant tissue housed in the metallic coating.

**Figure 7 sensors-24-02160-f007:**
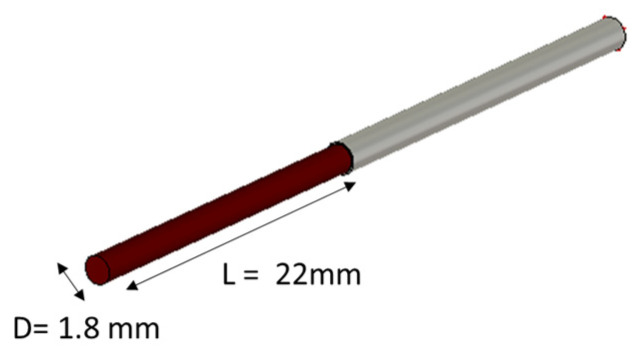
A graphical 3D CST model of the numerical liver malignant tissue placed in direct contact with the probe-to-tissue aperture’s plane in free space.

**Figure 8 sensors-24-02160-f008:**
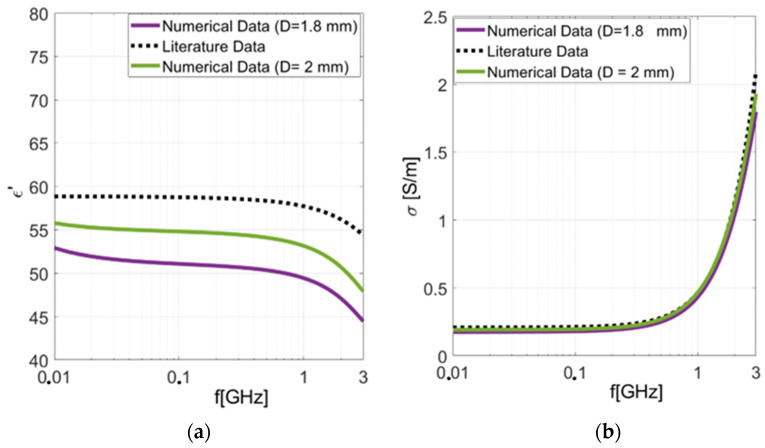
Dielectric properties computed for the two numerical cylindrical malignant liver tissues of different sizes compared with those of the literature’s dispersive model of the malignant liver [9]: (**a**) real part of permittivity; (**b**) conductivity.

**Figure 9 sensors-24-02160-f009:**
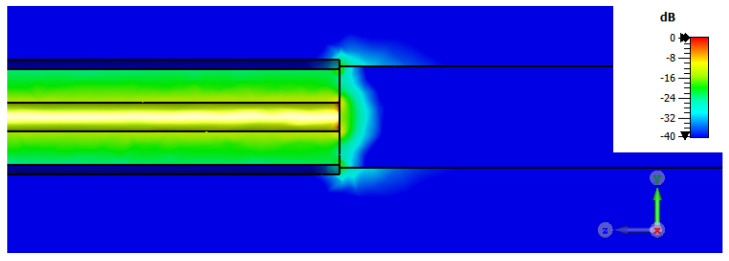
Electric field distribution when numerical liver malignant tissue is placed in direct contact with the probe-to-tissue aperture’s plane in free space.

**Figure 10 sensors-24-02160-f010:**
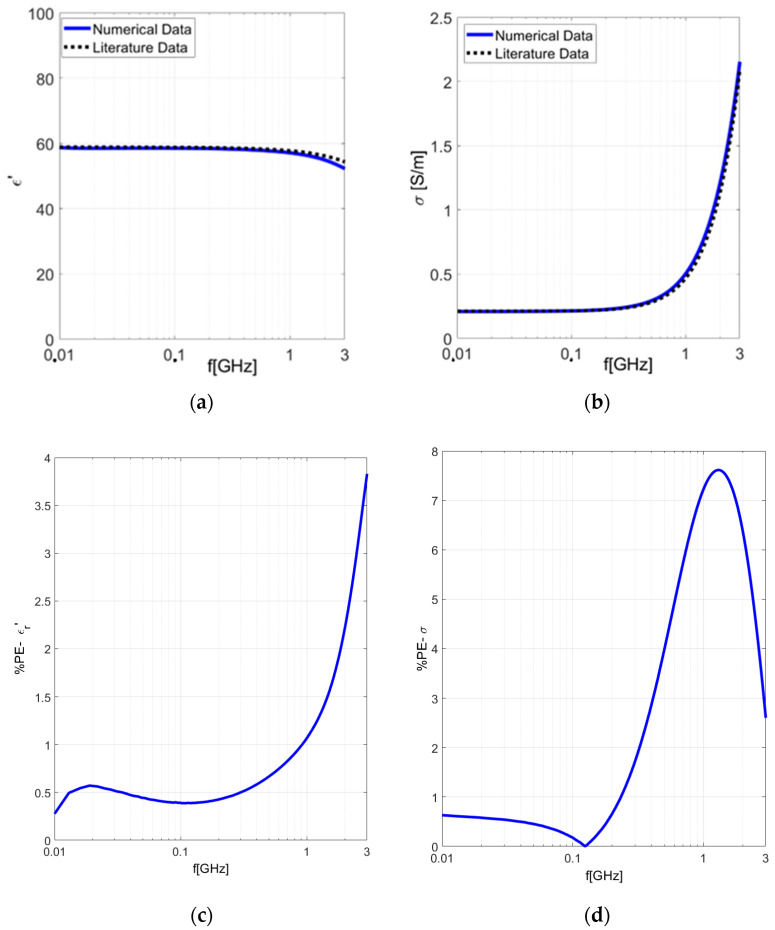
Numerical simulations: dielectric properties computed for the cylindrical numerical tissue of diameter D = 1.8 mm enclosed within a metal coating. (**a**) Real part of the permittivity of the malignant liver tissue. (**b**) Conductivity of the malignant liver tissue. (**c**) Percentage error (*PE*) calculated for the real part of the dielectric permittivity. (**d**) Percentage error (*PE*) calculated for the conductivity.

**Figure 11 sensors-24-02160-f011:**
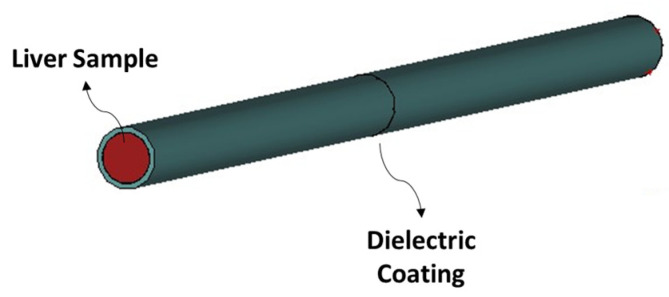
Pictorial 3D numerical design of a Teflon-coated open-ended coaxial probe.

**Figure 12 sensors-24-02160-f012:**
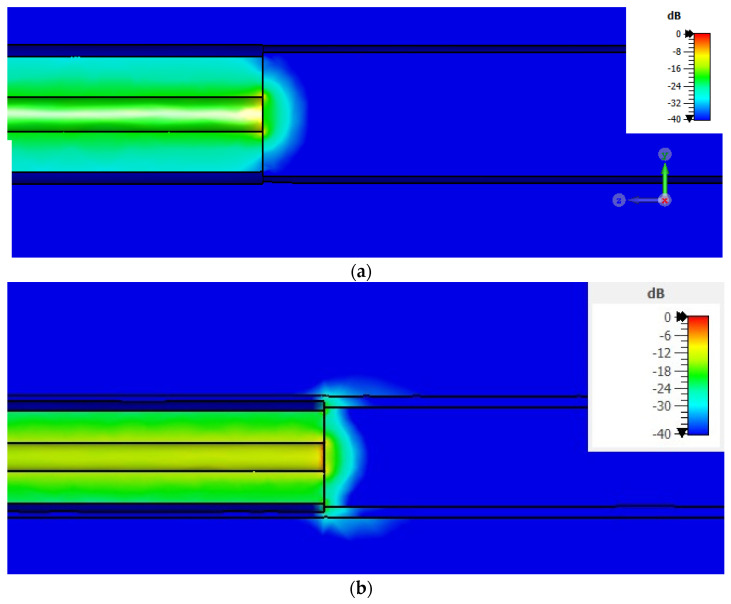
(**a**) Electric field distribution when numerical liver malignant tissue is placed inside the metallic coating; (**b**) electric field distribution when numerical liver malignant tissue is placed inside the dielectric coating.

**Figure 13 sensors-24-02160-f013:**
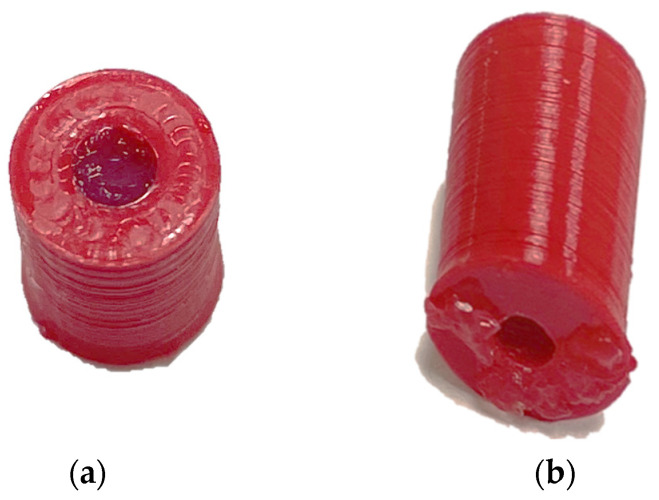
A 3D-printed cylindrical coating support made of PLA: (**a**) top view; (**b**) perspective view.

**Figure 14 sensors-24-02160-f014:**
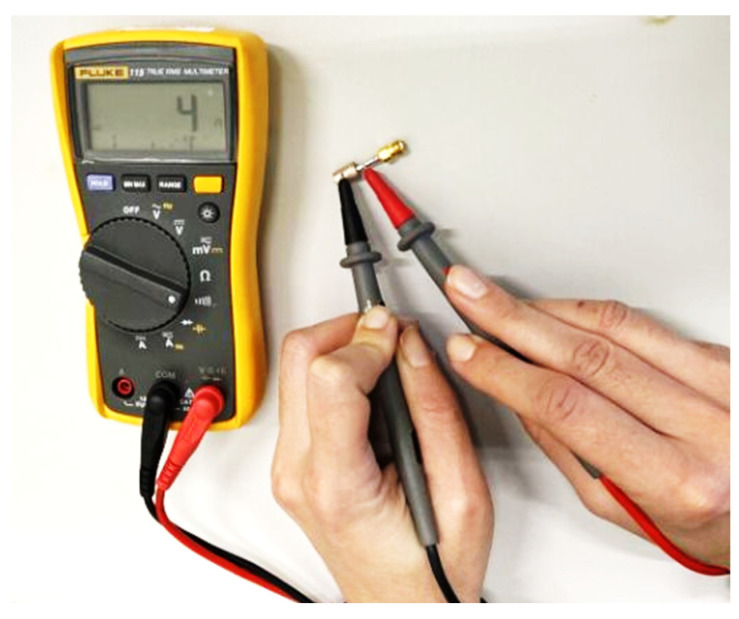
Laboratory test employing a digital multimeter to assess the electrical continuity between the metalized cylinder and the open-ended coaxial probe.

**Figure 15 sensors-24-02160-f015:**
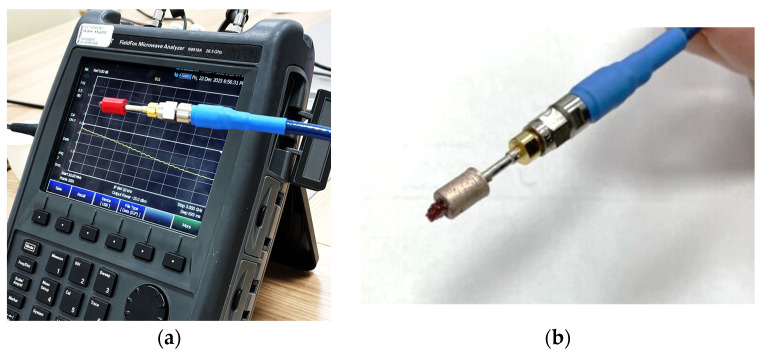
Measurement equipment consisting of calibrated VNA and open-ended coaxial probe containing bovine liver sample: (**a**) measurement setup with dielectric coating; (**b**) measurement setup with metalized coating.

**Figure 16 sensors-24-02160-f016:**
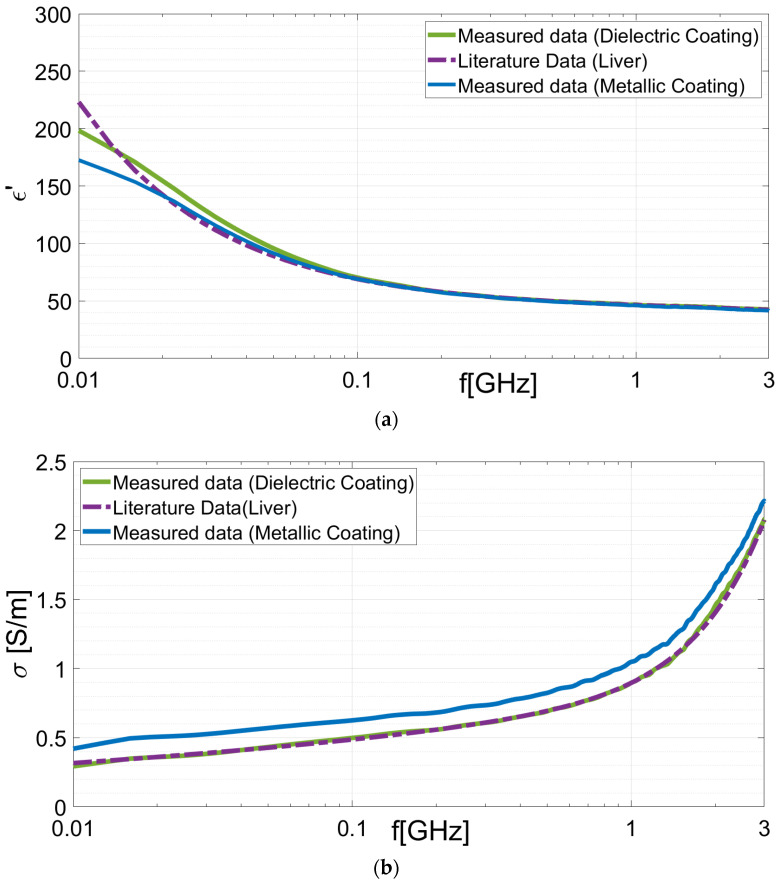
Experimental tissues: dielectric properties of the bovine liver placed inside the dielectric and metallic coatings. Dielectric constant (**a**) and conductivity (**b**) compared with those of the reference literature’s dispersive model.

**Table 1 sensors-24-02160-t001:** *PE* (percentage error) at 1 GHz in the cases of the two numerical cylindrical livers.

Test Case	PE (ε′)	PE (σ)
Numerical Cylindrical Liver D = 1.8 mm	14.43%	7.28%
Numerical Cylindrical Liver D = 2 mm	8.6%	2.1%

**Table 2 sensors-24-02160-t002:** *PE*s (percentage errors) at 1 GHz in the case of the 1.8 numerical cylindrical tissue hosted in metallic and dielectric coatings.

Test Case	PE (ε′)	PE (σ)
1.8 mm Numerical Liver: Metallic Coating	1%	7.2%
1.8 mm Numerical Liver: Dielectric Coating	7.8%	0.66%

**Table 3 sensors-24-02160-t003:** PEs (percentage errors) at 1 GHz, as obtained from the measurements of the bovine liver sample hosted in metallic and dielectric coatings.

Test Case	PE (ε′)	PE (σ)
Experimental Liver: Metallic Coating	0.89%	12.15%
Experimental Liver: Dielectric Coating	0.88%	0.068%

## Data Availability

Data are contained within the article.

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
