# Peer review of "Open-Ended Coaxial Probe for Effective Reconstruction of Biopsy-Excised Tissues’ Dielectric Properties"

_sensors, 2024, doi:10.3390/s24072160_

Round 1

Reviewer 1 Report

Comments and Suggestions for Authors

This article reported a method to improve the accuracy of dielectric characterization of millimeter-scale biopsies. Coated open coaxial probes with dielectric coatings and metallic coating used for dielectric characterization of biopsy tissue samples are compared. The obtained results are instructive for practical applications, but there are still some problems at the same time, can be published with major modifications. 

1. What is the metal used in the metal coating at the simulation and experimental section? Why was that metal chosen as the coating? Would other metals have worked better? A detailed description needs to be given.

2. In the experimental validation section, the authors used a PLA cylinder instead of the dielectric PTFE coating, so why not just use a Teflon coating? Please explain the reason.

3. The numerical results given in 3.1 show that metallic coatings are far more accurate than dielectric coatings for testing dielectric properties, however, this advantage is not apparent in the experimental results of 3.2. This might be the main point of contradiction in this article. Although the authors explain this, it is far from sufficient. The tests data of other biopsy samples are recommended to supplemented.

4. Is there any solution to the problem of the peeling of metallic paint and uneven distribution of metallic coating? It would be better to address these disturbances in the experimental verification section before giving strong comparative data.

Author Response

We greatly appreciated the Reviewer's time, careful examination of the manuscript and the valuable suggestions provided.

Please find the detailed responses below and the corresponding revisions/corrections in red in the re-submitted files.

Reviewer 2 Report

Comments and Suggestions for Authors

This reports on the use on an open-ended coaxial probe for the measurement of dielectric values of mm-sized biopsies of tissue, especially liver, in the range 0.01 – 3 GHz. The aim appears to be to eventually differentiate normal from malignant tissue. The work is obviously useful, but the paper needs to be improved in a number of ways as follows:

·       The method seems to rely on a previously-developed algorithm for extracting permittivity and conductivity from reflection coefficients. The method appears to have been presented in a number of conference presentations (Refs, 25, 32 – 34) which give only the briefest of details. Even if the method is identical to one of those described in Berube et al (Ref 36) it would help the reader to include a brief description of the approach and the assumptions made.

·       In places, it is not clear whether the data refers to actual tissue or numerical simulation (Fig 1 vs Fig 3; Fig 9 vs Fig 15). Suggest start caption with ‘Numerical simulation:’ or ‘Experimental tissue:’ On this, I note that Fig 7, 9 are of numerical simulation of malignant liver, whereas Fig 15 is presumably normal ‘real’ tissue. Is there a comparison anywhere between numerical simulation of normal liver with malignant liver simulations, to make a case that there is sufficient difference between the two? Certainly Fig 15 shows a big difference in permittivity across the range. Please add to discussion.

·       There is also potential confusion in terms when referring to ‘Phantoms’. To many, a phantom refers to a physical object, such as a saline-filled vial designed to simulate the dielectric properties of tissue, rather than a numerical voxel-based model. Please make the distinction clear, especially since p4 - 6 refers to actual saline etc.

·       This may seem a minor point, but the source of actual liver samples needs to be given and whether an exemption from the usual requirement for animal ethics approval was granted. I am assuming that the liver was obtained from a retail outlet, but this raises additional questions of how long post-mortem was the sample obtained and whether post-mortem changes were considered.

·       Dielectric values are quite temperature dependent: has consideration been given to any adjustments which would need to be made in the case of human tissue biopsies (the present experiments appear to be conducted at room temperature).

·       Please provide a few more details of the parameters used the CST modelling e.g voxel size, whether adaptive gridding used etc. What is the input function?

·       How is the tissue inserted into the needle-shaped probe so that no air remains at the end?

Comments on the Quality of English Language

Mostly fine. Find a different word to 'delved into' at line 223 (rather colloquial) 'investigated' better. Also 'turned' zero line 130 ('became')

Author Response

(The authors gave the same response as above.)

Round 2

Reviewer 1 Report

Comments and Suggestions for Authors

The paper has been fully revised and can be accepted for publish.